# Analysis of Preventable Risk Factors for *Toxoplasma gondii* Infection in Pregnant Women: Case-Control Study

**DOI:** 10.3390/jcm11041105

**Published:** 2022-02-19

**Authors:** Carlo Bieńkowski, Małgorzata Aniszewska, Monika Kowalczyk, Jolanta Popielska, Konrad Zawadka, Agnieszka Ołdakowska, Maria Pokorska-Śpiewak

**Affiliations:** 1Doctoral School, Medical University of Warsaw, Żwirki i Wigury 61, 02-091 Warsaw, Poland; 2Department of Children’s Infectious Diseases, Medical University of Warsaw, Wolska 37, 02-091 Warsaw, Poland; malgorzata.aniszewska@wum.edu.pl (M.A.); jolanta.popielska@wum.edu.pl (J.P.); konrad.zawadka@wum.edu.pl (K.Z.); agnieszka.oldakowska@wum.edu.pl (A.O.); mpspiewak@gmail.com (M.P.-Ś.); 3Hospital of Infectious Diseases, 01-201 Warsaw, Poland; 4Department of Epidemiology of Infectious Diseases and Surveillance, National Institute of Public Health NIH—National Research Institute, 00-791 Warsaw, Poland; mkkowalczyk1@gmail.com

**Keywords:** toxoplasmosis, pregnancy, congenital infection, zoonosis

## Abstract

Background: *Toxoplasma gondii* (TG) is a parasitic protozoon that may cause miscarriages or birth defects if the infection occurs during pregnancy. The study’s aim was to evaluate the risk factors associated with TG infection in pregnant women. Materials: Medical charts for all 273 pregnant women with suspected TG infection consecutively admitted to the Hospital of Warsaw between 2019 and 2020 were retrospectively analyzed. The presumptive TG diagnosis was verified by a serologic assessment of IgM and IgG titers, and IgG affinity tests. Results: The median age was 32 years (range: 19–42 years). The diagnosis of primary TG infection was confirmed in 74/273 (27.1%) women. In 114/273 (41.8%) there was evidence of past infection. In 71/273 (26%) women, an infection was excluded. In 172/273 (62%) women the recommended testing for other infectious diseases putting fetus development at risk was performed correctly. Logistic regression model analysis revealed that living in rural areas and eating raw meat were independent factors associated with increased risk of TG infection during pregnancy (OR 2.89, 95% CI: 1.42–5.9, *p* = 0.004; and OR 2.07, 95% CI: 1.03–4.18, *p* = 0.04, respectively). Conclusions: The independent risk factors for TG infection during pregnancy include living in rural areas and eating raw meat. The physician’s educational role here is crucial for the efficient prevention of congenital toxoplasmosis.

## 1. Introduction

*Toxoplasma gondii* (TG) is a parasitic protozoon. Its life cycle includes sexual reproduction in the final hosts (felids), where epithelial cells of the small intestine are infected; following the infestation, oocysts are then excreted with the feces [1]. Under appropriate environmental conditions, oocysts transform into sporocysts and can maintain their invasive potential even for several years. Vegetative reproduction takes place in intermediate hosts (humans, other mammals, and birds) where TG is spread by lymphatic and blood vessels (in monocytes and granulocytes) to distant organs and tissues. Tachyzoites form in large numbers and destroy the host cells, and then, in people with properly functioning immune systems, they are converted into resting forms—tissue cysts [2,3].

The two major routes of infection in humans include oral (via contaminated meat, hands, soil, cat feces) and transplacental transmission, resulting in acquired or congenital infection, respectively. Rare instances of transmission include organ transplant recipients who get infected by receiving organs from *Toxoplasma*-positive donors. The incidence of congenital toxoplasmosis is estimated at 1–4/1000 newborns [4].

The course of the infection in adults is usually asymptomatic or mild. However, it may pose a risk for the fetus’s development of abnormalities if a pregnant woman becomes infected. The risk of transplacental infection increases with the duration of pregnancy. However, the consequences for the fetus are most severe when an infestation occurs during the first months of pregnancy. In the first trimester, the probability of TG transmission is lowest and estimated at 17–25%, however, if the parasite is transmitted, this often leads to miscarriage [4]. In the second and third trimesters of pregnancy, the likelihood of tachyzoites infesting the fetus increases (25–50% and 60–90%, respectively), and as a consequence, severe abnormalities, such as hydrocephalus, intracerebral calcification or sight damage may occur [4].

The aim of this study was to evaluate the potential risk factors associated with TG infection in pregnant women.

## 2. Material and Methods

The authors retrospectively analyzed the medical charts of all pregnant women, referred by their gynecologists to the Regional Hospital of Infectious Diseases in Warsaw due to suspected TG infection (based on a positive serologic assessment of anti-TG antibody titers) between 1 September 2019 and 14 March 2020. All patients in the above category referred to the hospital within these dates, in the order they were referred, were included in the study. The final study group consisted of women with a confirmed primary *Toxoplasma gondii* infection, while the control group consisted of women for whom the infection had been excluded. Women with evidence of past infection and women with inconclusive results were excluded from the final analysis.

The analysis included anamnesis data on potential risk factors for TG infection, such as age, socioeconomic status, history of miscarriage, caring for domestic and/or wild cats currently or in the past, long-distance travels, gardening without gloves, eating unwashed vegetables, or eating raw meat currently or in the past. In addition, implementation of screening testing towards other infections, including human immunodeficiency virus (HIV), hepatitis B virus (HBV), hepatitis C virus (HCV), cytomegalovirus (CMV), syphilis, and rubella were also included in the analysis. The presumptive diagnosis of TG infection was verified by a serologic assessment of immunoglobulin M (IgM) and immunoglobulin G (IgG) titers, and IgG affinity tests using Enzyme-Linked Fluorescent Assay (ELFA) by VIDAS^®^ (bioMérieux, Lyon, France). The diagnostic algorithm for TG infection evaluation is presented in Figure 1.

An infection was excluded when both IgM and IgG antibody titers were negative. When IgM was positive and IgG negative, a primary infection was possible and another assessment was needed after 1–3 weeks. When both IgM and IgG were positive, an infection was suspected and an IgG affinity assessment was required. When IgM was negative and IgG positive, an infection had possibly occurred in the past, and an IgG affinity assessment was necessary. Low affinity indicated a primary infection, high affinity revealed an infection that had occurred in the past, while other results needed had to be reassessed. All women diagnosed with toxoplasmosis were referred for amniocentesis, but as no further observation was performed, there were no data confirming the congenital toxoplasmosis. The testing scheme for vertical infections was considered correctly implemented if the first testing towards toxoplasmosis, rubella, HIV, HCV, and syphilis were performed before the tenth week of gestation. If the first tests for TG were negative, reassessment was recommended between the twenty-first and twenty-sixth week of gestation. Moreover, testing for HBV and HIV was recommended between the thirty-third and thirty-seventh week of gestation (Table 1).

## 3. Statistical Analysis

The normality of continuous variables was tested using the Shapiro–Wilk’s test. The Mann–Whitney U test was used to compare continuous variables and the Chi^2^ test was used to evaluate categorical variables. A *p*-value of <0.05 was considered significant. Multivariate analysis was performed using a logistic regression model, where candidate predictors were entered into the model irrespective of the results of the univariate analysis. After entering all variables into the model, the variables that showed the least significant associations were subsequently excluded until all variables remained significant (*p* < 0.05). Statistical analysis was performed with Medcalc ver. 20.009, Ostend, Belgium.

### Ethical Statement

The design of the work conforms to standards currently applied in the Medical University of Warsaw’s Bioethics Committee. Approval number: AKBE/132/2021.

## 4. Results

The medical records of 273 pregnant women with suspected TG infections were analyzed. The median age was 32 years (range: 19–42 years). In 119/273 (43.6%) of the participants, the place of residence was in a rural area, 44/273 (16.1%) had a history of miscarriage, and 21/273 (7.7%) had a history of long-distance travel. Chronic diseases were reported in 69/273 (25.3%) of pregnant women and 53/273 (19.4%) had autoimmune diseases.

Women with confirmed toxoplasmosis were younger than the women in the control group (28 years (IQR: 24–32 years) vs. 32 years (IQR 29–35 years), *p* < 0.001). Moreover, women with a TG infection were more likely to live in rural areas (55.4% vs. 28.2%, *p* < 0.001), more often ate raw meat before their pregnancy (58.1% vs. 38.0%, *p* = 0.016), and more often gave care to cats during pregnancy (35.1% vs. 16.9%, *p* = 0.01) (Table 2). Multivariate logistic regression revealed that living in a rural area (OR 2.89, 95% CI 1.42–5.9, *p* = 0.004), and eating raw meat (OR 2.07, 95% CI: 1.03–4.18, *p* = 0.04) were independent risk factors for TG infection during pregnancy (Table 3).

The diagnosis of primary TG infection was confirmed in 74/273 (27.1%) women, who were then treated with spiramycin. In 114/273 (41.8%) women, there was evidence of past infection. In 71/273 (26%) women, infection was excluded. The remaining women (14/273, 5.1%) had inconclusive results, and reassessment was recommended.

The clinical evaluation of the pregnant women did not reveal any significant differences between women with confirmed TG infection when compared to the control group regarding fetal ultrasound results (91.9% vs. 93%, *p* = 0.8), and the presence of lymphadenopathy (8.1% vs. 2.8%, *p* = 0.16), the presence of influenza-like symptoms (16.2% vs. 11.3%, *p* = 0.4), or both symptoms combined together (5.4% vs. 1.4%, *p* = 0.19).

In 172/273 (62%) women, the recommended testing procedures for other infectious diseases dangerous for the fetus’s development were carried out correctly.

## 5. Discussion

The risk of transplacental TG infection increases with the duration of pregnancy. However, the consequences for the fetus are most severe when an infestation occurs in the first months of pregnancy. Therefore, knowledge about TG risk factors seems to have an influence on the prevention of TG infection during pregnancy [3]. However, Serdarian et al., who analyzed 653 pregnant women with IgG detected in their serum and the B1 gene of *T. gondii* found in their placental tissue using a nested-PCR assay, concluded that the detection of the B1 gene in placental tissues of the healthy newborn infants reiterates that the presence of *T. gondii* in the placenta does not always result in congenital toxoplasmosis [5].

Ferguson et al. investigated a cohort of mothers who vertically infected their children, and they revealed that 73% of the women lacked knowledge concerning the risk factors for TG infection or its potential threat to the fetus [6]. Therefore, the analysis of preventable risk factors for TG infection during pregnancy seems to be important in this matter. As in our cohort, dwelling place and eating habits increased the risk of infection during pregnancy by 2.89 times and 2.07 times, respectively.

Awareness of local seroprevalence trends, particularly in the women of childbearing age, may allow proper public health policies to be applied, targeting, in particular, seronegative women of childbearing age in high seroprevalence areas [7]. Rostami et al., in their meta-analysis, investigated geo-climatic factors and the prevalence of chronic toxoplasmosis in pregnant women. They concluded that different regions of the world may benefit from different types of interventions, and thus, novel preventive measures for a region should be developed according to local climate, agricultural activities, and the peoples’ cultural attitudes [8]. However, no studies to date have included living in rural areas as a potential risk factor for TG infection.

In our analysis, older women were more likely to have a TG infection excluded (they were seronegative), which is contrary to results obtained by Fanigiulio et al., where seropositivity was more common in older women [9]. Further analyses in this matter are needed.

Tarekegn et al., in a systematic review and meta-analysis of the potential risk factors associated with seropositivity for TG among pregnant women and HIV-infected individuals, analyzed 24 reports, which included 6003 individuals (4356 pregnant women and 1647 HIV-infected individuals). They concluded that a significant overall effect of anti-*Toxoplasma gondii* seropositivity among pregnant women (*p* < 0.05) was associated with age, abortion history, contact with cats, cat ownership, having knowledge about toxoplasmosis, being a housewife, and having unsafe water sources [10].

In our analysis, the factors that independently increase TG infection risk among the population of pregnant women in our region include living in rural areas and eating raw meat. Living in the countryside may be influenced by a lot of factors and may be difficult to change. However, avoiding raw meat in the daily diet can be implemented by women planning a pregnancy or while pregnant.

Cerro et al. investigated the seroprevalence of TG among cats in Peru and revealed that it is associated with a cat’s eating habits. Those fed with raw meat were more exposed compared to those fed with commercial cat food (chi^2^ = 9.50, *p* = 0.004) or with homemade food (chi^2^= 4.1, *p* = 0.027) [11]. In our study, caring for cats was significantly associated with TG infection during pregnancy, but there was no difference between domestic cats and wild cats. Hence, we cannot be certain of these cats’ eating habits. Moreover, Cerro et al. showed that 88% of cats were diagnosed with the chronic phase of TG infection; [11] therefore, every unknown cat, whether living at home or outdoors, poses a potential threat to previously uninfected women from TG infection.

Wei et al., in their systematic review and meta-analysis of the efficacy of anti-TG medicines in humans, concluded that the risk of vertical transmission of TG was approximately 9.9% when the infected mother received the necessary treatment with spiramycin alone or combined with pyrimethamine-sulfadiazine, against *T*. *gondii* [12]. However, Thiebaut et al., in their meta-analysis of the effectiveness of prenatal treatment for congenital toxoplasmosis, found only weak evidence of an association between early treatment and a reduced risk of congenital toxoplasmosis [13]. In addition, we revealed that 74/273 (27.1%) participants in our cohort had their diagnoses confirmed, and they were treated accordingly (with spiramycin). However, we lost the patients to follow-up (our hospital was transformed into a COVID-19-only hospital), therefore, we could not perform further analysis on the treatment’s efficacy.

Wallon et al. proved that introducing monthly prenatal screening and improving antenatal diagnosis was associated with a significant reduction in the rate of congenital infection [14]. In our study, only 62% of women had the recommended testing scheme carried out correctly, which included performing tests for TG according to the Polish recommendations presented in Table 1.

The limitations of our study include a non-detailed differentiation of potential risk factors for TG infection, and there are no time frames for the occurrence of some of the risk factors. The data were collected from medical records retrospectively, and the medical records were prepared by more than one physician. However, for finding a cause and effect relationship between TG infection during pregnancy and its risk factors, a case-control study is methodologically justified.

To conclude, the independent risk factors for TG infection in pregnancy in our region include living in rural areas and eating raw meat. Twenty-six percent of women who were suspected of having toxoplasmosis by their gynecologists were not infected. Only 62% of women had the recommended testing scheme carried out correctly. The educational role of a physician in these matters is crucial for the effective prevention of congenital toxoplasmosis. Further studies are recommended to deepen the analysis of important risk factors for TG during pregnancy in order to support the development of more cost-effective preventive strategies.

## Figures and Tables

**Figure 1 jcm-11-01105-f001:**
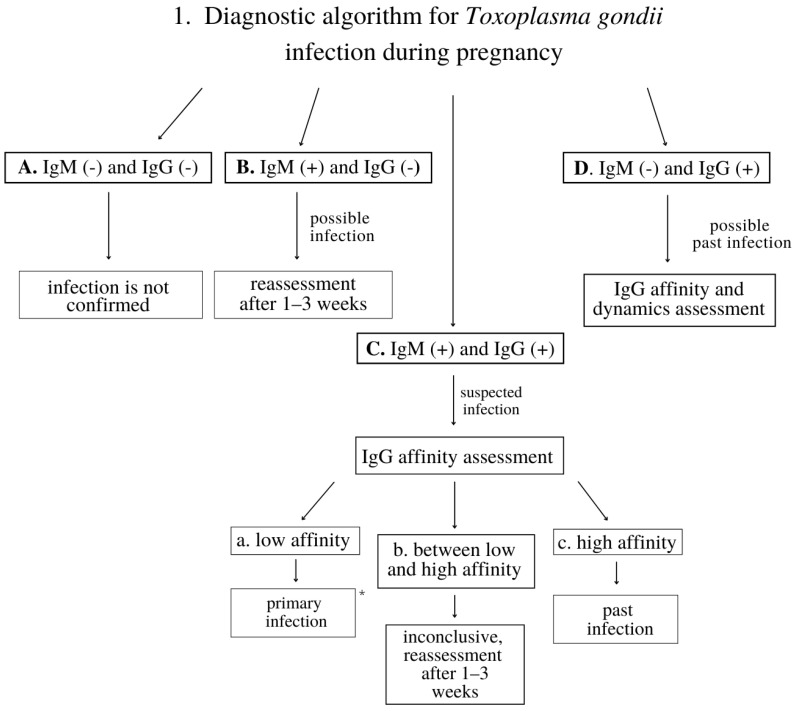
The diagnostic algorithm for *Toxoplasma gondii* infection during pregnancy. * All women diagnosed with toxoplasmosis were referred for amniocentesis, but no further observation was carried out.

**Table 1 jcm-11-01105-t001:** Recommendations concerning diagnostic testing for infectious diseases during pregnancy according to the Polish Journal of Laws.

Examination Date	Diagnostic Tests
Up to the 10th week of gestation or at the time of first reporting	VDRL test. * Human immunodeficiency virus (HIV )and Hepatitis C virus (HCV) testing. Testing for toxoplasmosis (IgG **, IgM ***) unless the pregnant woman shows a result confirming the presence of IgG antibodies from before pregnancy. Rubella test (IgG, IgM), if the pregnant woman has not been ill or has not been vaccinated or in the absence of information.
Week 21–26th of gestation	In women with negative results in the first trimester—testing for toxoplasmosis (IgM).
Week 33–37th of gestation	Testing the HBs **** antigen presence. HIV testing. Vaginal and rectal culture for B-hemolytic streptococci (weeks 35–37 of gestation). VDRL and HCV studies in a group of women with an increased risk of infection.

* VDRL—venereal disease research laboratory (testing for syphilis); ** IgG—immunoglobulin G; *** IgM—immunoglobulin M; **** HBs—hepatitis B virus antigen.

**Table 2 jcm-11-01105-t002:** Baseline characteristics and clinical data on women with a confirmed diagnosis of primary toxoplasmosis compared to the control group of women where the infection was excluded.

Characteristic	Total *n* = 273	TG+ * *n* = 74	TG− ** *n* = 71	*p*-Value
Age in years, median (IQR) ****	30 (26–33)	28 (24–32)	32 (29–35)	<0.001
Living in rural areas, *n* (%)	119 (43.6)	41 (55.4)	20 (28.2)	<0.001
History of miscarriage, *n* (%)	44 (16.1)	12 (16.2)	15 (21.1)	0.45
Good socioeconomic status, *n* (%)	267 (97.8)	72(97.3)	71 (100)	0.58
Long-distance travels, *n* (%)	21 (7.7)	4 (5.4)	5 (7)	0.68
Chronic diseases, *n* (%)	69 (25.3)	20 (27)	22 (31)	0.6
Autoimmune diseases, *n* (%)	53 (19.4)	15 (20.3)	18 (25.4)	0.5
Gardening without gloves, *n* (%)	29 (10.6)	7 (9.5)	2 (2.8)	0.1
**Eating habits**				
• unwashed vegetables, *n* (%)	34 (12.5)	7 (9.5)	4 (5.6)	0.38
• raw meat before pregnancy, *n* (%)	150 (55)	43 (58.1)	27 (38)	0.016
• raw meat during pregnancy, *n* (%)	55 (20.1)	17 (23)	9 (12.7)	0.1
**Caring for cats**				
• during pregnancy, *n* (%)	91 (33.3)	26 (35.1)	12 (16.9)	0.01
• in the past, *n* (%)	106 (38.8)	30 (40.5)	19 (26.8)	0.08
• domestic cats, *n* (%)	96 (35.2)	25 (33.8)	13 (18.3)	0.03
• wild cats, *n* (%)	80 (29.3)	23 (31.1)	12 (16.9)	0.046
Correctly implemented testing for other infectious diseases ***, *n* (%)	172 (63)	45 (60.8)	51 (71.8)	0.16
**Clinical evaluation**				
Correct ultrasound result, *n* (%)	251 (91.9)	68 (91.9)	66 (93)	0.8
Lymphadenopathy, *n* (%)	14 (5.1)	6 (8.1)	2 (2.8)	0.16
Influenza-like symptoms, *n* (%)	35 (12.8)	12 (16.2)	8 (11.3)	0.4
Both lymphadenopathy and influenza-like symptoms, *n* (%)	41 (15)	4 (5.4)	1 (1.4)	0.19

* TG+—primary toxoplasmosis; ** TG−—excluded toxoplasmosis.; *** Women who had been correctly tested according to Polish recommendations (presented in Table 1). **** IQR—interquartile range.

**Table 3 jcm-11-01105-t003:** Univariate and Multivariate logistic regression analyses of factors associated with primary *Toxoplasma gondii* infection.

	Univariate			Multivariate		
Factor	Odds Ratio	95% Confidence Interval	*p*-Value	Odds Ratio	95% Confidence Interval	*p*-Value
Living in rural areas	3.17	1.59–6.32	0.001	2.89	1.42–5.90	0.004
Eating raw meat	2.25	1.59–4.38	0.017	2.07	1.03–4.18	0.04
Caring for wild cats	2.22	1.00–4.90	0.049	1.72	0.72–4.10	0.22
Caring for domestic cats	2.27	1.05–4.92	0.03	1.83	0.79–4.27	0.16

Data are presented as odds ratio (95% CI), *p*-value. Candidate predictors were entered into the model irrespective of the results of the univariate analysis. After entering all variables into the model, the variables that showed least significant associations were subsequently excluded until all variables remained significant (*p* < 0.05).

## Data Availability

Available on reasonable request to the Corresponding Author.

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
