# Peer review of "Analysis of Preventable Risk Factors for Toxoplasma gondii Infection in Pregnant Women: Case-Control Study"

_jcm, 2022, doi:10.3390/jcm11041105_

Round 1

Reviewer 1 Report

This is an interesting study of the contribution of T gondii to infection in pregnant women in Poland.

There could be some improvement in the language, get a professional English native speaker to review the text before re-submission.

In the Introduction it could be made clearer that primary/first infection during pregnancy is thought to be bringing about infection of the fetus, then clarify that this criterion (IgM positivity) is presumably used in M&Ms to arrive at the 74 cases. That could be made clearer. It might be also useful to add a comment about the total number of Tg seropositive pregnant females (presumably 74 + 114?), and compare that to the literature. Did you compare all Tg positives (primary and chronic) with the control group - one might assume that the two major risk factors would actually stay the same, regardless of recency of infection.

Cat contact is initially statistically associated with positivity - but drops out after further analyses, might still be worth discussing? Not sure that the Cerro study from Peru is a useful citation - we are not worried about cats' eating habits?

I guess you could tell pregnant women to stop consuming raw meat, but can they stop living where they live?

The focus of the paper are risk factors, not treatment - so I am not convinced that the two references to treatment papers (Wei and Thiebaut) are warranted.

Author Response

Please find my response in the attached file.

Reviewer 2 Report

In the present study, medical charts of consecutive 273 pregnant women with suspected Toxoplasma gondii infection admitted to the Warsaw’s hospital were retrospectively analysed. The presumptive TG diagnosis was verified using serologic assessment of IgM, IgG titters, and IgG affinity tests. It was shown that main risk factors for Toxoplasma gondii infection in pregnancy are living in rural area and eating raw meat. The significantly of the study is at the regional level. The manuscript could be accepted after answer to certain questions and needed corrections. Dear authors, please respect the journal and at least try to make a correct reference list…

Remarks

  1. Abstract: based on journal’ requirements “the abstract should be a total of about 200 words maximum.” Now it is 295 words. A lot of not essential things are presented in abstract. For instance, please see my suggestion and correct it “Background: Toxoplasma gondii (TG) is a parasitic protozoon that may cause miscarriages or birth defects if the infection occurs during pregnancy. The aim of the study was to evaluate the risk factors for TG infection in pregnant women. Materials: Medical charts of consecutive 273 pregnant women with suspected TG infection admitted to the hospital of Warsaw between 2019 till 2020 were retrospectively analysed. The presumptive TG diagnosis was verified using serologic assessment of IgM, IgG titters, and IgG affinity tests. Results: The diagnosis of primary TG infection was confirmed in 74/273 (27.1%) women. In 114/273 (41.8%) there was evidence of past infection. In 71/273 (26%) women the infection was excluded. In 172/273 (62%) women the recommended testing towards other infectious diseases posing risk for the foetus development was carried out correctly. Logistic regression model analysis revealed living in rural area and eating raw meat as independent factors associated with increased risk of TG infection during pregnancy (OR 2.89, 95% CI: 1.42-5.9, p=0.004, and OR 2.07, 95% CI: 1.03-4.18, p=0.04, respectively). Conclusions: Independent risk factors for TG infection in pregnancy include living in rural area and eating raw. Educational role of a physician in these matters is crucial for efficient prevention of congenital toxoplasmosis.”
  2. “with suspected TG infection” (Abstract and Methods) it is not clear by which signs/test infection was suspected, state it clearly in methods.
  3. Toxoplasma gondii. Throughout manuscript it should be in italic!!!
  4. Keywords are missing. Based on journal requirements “Keywords:Three to ten pertinent keywords need to be added after the abstract. We recommend that the keywords are specific to the article, yet reasonably common within the subject discipline”.
  5. “they are converted into resting forms – bradyzoites” change to … – tissue cysts.
  6. “In addition, assessment of the correct implementation of recommended screening testing towards other congenital infections was also performed” it is not clear how this aim is related to Toxoplasma infection in pregnancy. I do not agree to raise this point as the aim of the study.
  7. Methods: improve this section, more details are needed. “In order to minimize the study bias, women with the evidence of past infection and women with in-conclusive results were excluded from the final analysis” not clear, data/numbers are needed. “eating imported meat” not clear, how it is related with infection local/imported meat, such data are not incorporated in Tables 2 and 3.
  8. Results: “had a good socioeconomic status” how have you evaluated this? Arbitrary. I suggest to not include this factor. “Women with confirmed toxoplasmosis were younger than women in the control group (28 years [IQR: 24-32 years] vs. 32 years [IQR 29-35 years], p<0.001).” how can you explain these findings? Please include explanation in discussion section. Table 2 “Correctly implemented testing for other infectious diseases, n (%)” needs clarification in the footnote of the table. „Moreover, women with TG infection were more likely to live in rural area (55.4% vs. 28.2%, p<0.001), more often ate raw meat (59.5% vs. 39.4%, p=0.016), and more often were giving care to cats (48.7% vs. 30.1%, p=0.03) (Table 2).“ The reliability of the data on eating raw meat before and during pregnancy differs. I do not known whether it is correct to say that eating raw meat is significantly different, because authors compared the overall indicator.
  9. Discussion: please emphasize what was novelty of your study? What were differences from other studies? or it is just repetitions of similar studies carried out in other countries. “Wallon et al. proved that introduction of monthly prenatal screening and improve-ment in antenatal diagnosis were associated with a significant reduction in the rate of congenital infection. [13]. In our study, only 62% of women had the recommended testing scheme carried out correctly.” Not clear how it is related with Toxoplasma study?
  10. References: have you read requirements?

“References •  Journal Articles:

 Author 1, A.B.; Author 2, C.D. Title of the article. Abbreviated Journal Name Year, Volume, page range.

  • Books and Book Chapters:

 Author 1, A.; Author 2, B. Book Title, 3rd ed.; Publisher: Publisher Location, Country, Year; pp. 154–196.

 Author 1, A.; Author 2, B. Title of the chapter. In Book Title, 2nd ed.; Editor 1, A., Editor 2, B., Eds.; Publisher: Publisher Location, Country, Year; Volume 3, pp. 154–196.”

Author Response

Please find the response in the attached file.

Round 2

Reviewer 2 Report

I have no further remarks, authors have responded to the comments made by the reviewers. Nice study.